# Optoelectronic and Electrothermal Properties of Transparent Conductive Silver Nanowires Films

**DOI:** 10.3390/nano9060904

**Published:** 2019-06-21

**Authors:** Yuehui Wang, Dexi Du, Xing Yang, Xianfeng Zhang, Yuzhen Zhao

**Affiliations:** 1Zhongshan Institute, University of Electronic Science and Technology of China, Zhongshan 528402, China; dudexi_work@foxmail.com (D.D.); shirleywyh@126.com (X.Y.); 2School of Materials and Energy, University of Electronic Science and Technology of China, Chengdu 610054, China; 3Department of Materials Science and Engineering, Tsinghua University, Beijing 100084, China; zhaoyz@mail.tsinghua.edu.cn

**Keywords:** silver nanowire, optoelectronic properties, electrothermal properties, transparent conductive film

## Abstract

Silver nanowires (AgNWs) show promise for fabricating flexible transparent conductors owing to their excellent conductivity, high transparency, and good mechanical properties. Here, we present the fabrication of transparent films composed of AgNWs with diameters of 20–30 nm and lengths of 25–30 μm on polyethylene terephthalate substrates and glass slides substrates using the Meyer rod method. We systematically investigated the films’ optoelectronic and electrothermal properties. The morphology remained intact when heated at 25–150 °C and the AgNWs film showed high conductivity (17.6–14.3 Ω∙sq^−1^), excellent transmittance (93.9–91.8%) and low surface roughness values (11.2–14.7 nm). When used as a heater, the transparent AgNW conductive film showed rapid heating at low input voltages owing to a uniform heat distribution across the whole substrate surface. Additionally, the conductivity of the film decreased with increasing bending cycle numbers; however, the film still exhibited a good conductivity and heating performances after repeated bending.

## 1. Introduction

Transparent conductive films (TCFs) are one of the important parts of many optoelectronic devices such as touch panels, film heaters, and organic light-emitting diodes [1,2,3,4,5]. It is anticipated that TCFs will be applied as transparent film heaters (TFHs) in various applications, such as outdoor displays, vehicle window defrosters, or heat retaining windows [4,5,6,7,8,9,10]. Currently, indium tin oxide (ITO) is the most widely used TCF because of its outstanding optoelectronic properties compared with other materials [1,2,3,4]. However, its brittle ceramic properties and expensive vacuum deposition process limit its application in flexible TCFs. Meanwhile, ITO shows a slow temperature response owing to its intrinsic properties [2].

To replace ITO, several emerging materials have been developed, such as metal nanowires and meshes [6,7,8,9], graphene [3,4], Ga-doped ZnO [10], and carbon nanotubes [5]. However, some problems still need to be solved before those materials can be widely used for TFH application. Carbon nanotubes have large resistance and graphene’s fabrication process is highly complex [11]. 

In recent years, silver nanowires (AgNWs) have been developed and now rank among the most promising candidate for replacing ITO owing to their excellent optoelectronic properties and large-scale and cost-effective fabrication process [12,13,14,15,16]. However, it has been a challenge to achieve both excellent optical transmittance and high conductivity because these two properties follow opposing trends, and this often results in optoelectronic performances far inferior to those of ITO [17,18,19,20]. Additionally, it is hard to obtain uniformly interconnected large-area AgNWs networks, particularly when using solution processes. Randomly non-uniform AgNWs networks result in the discontinuous heating of TFH. The literature indicates that the underlying reasons for the limited performance of AgNW films may be related to the morphology of the AgNWs and the processes used to obtain high quality AgNW film [21,22,23,24,25,26]. However, experimental and theoretical studies have shown that AgNWs with high aspect ratios are better for forming the uniform conductive networks that improve the optoelectronic properties of AgNW film [27,28,29,30,31], which is related to the various models for electrical percolation [32,33].

To the best of our knowledge, there are few reports that systemically investigate the effects of temperature on the optoelectronic and electrothermal properties transparent conductive AgNW films. The reason for this may be because AgNW films are mainly used at low temperatures, especially when used as flexible TCFs. In this study, we fabricated transparent films composed of AgNWs with diameters of 20–30 nm and lengths of 25–30 μm on polyethylene terephthalate (PET) and glass slides substrates using the Meyer rod method and systematically investigated the optoelectronic and electrothermal properties of AgNW films. 

## 2. Experimental Approach

A 10 wt % solution of AgNWs (20–30 nm in diameter and 25–30 μm in length from Suzhou Gushi New Materials Co., Ltd., Suzhou, China) in isopropyl alcohol was diluted to 3.3 mg∙L^−1^. Then, the AgNWs solution was agitated in an ultrasonic bath for 5 min. Further, 1 mL AgNWs solution was placed onto a PET substrate (from Hefei Microcrystalline Materials Co., Ltd., Hefei, China) with dimensions of 210 mm × 297 mm or onto a glass slide substrate (from Jiangsu Shitai Experimental Equipment Co., Ltd., Shitai, China) with dimensions of 25 mm × 75 mm. Finally, a thin film was formed via the Meyer rod coating technique at a speed of 0.05 mm/s. Our coating process was performed at 26 °C and at a relative humidity of 50–60%. The wet films were air dried at room temperature for 2 min and then treated on digital hotplate at temperature between 25 and 250 °C for 20 min. After the films had cooled down to room temperature, we measured their optoelectronic properties and analyzed the films’ microstructures.

AgNWs-based film heater was made by attaching the two ends of the film to two electrodes (clips coated with copper foil). A direct current voltage was supplied by a power supply (GPD-3303s, Suzhou Guwei electronics Co., Ltd., Suzhou, China) to the film heater through those two clips coated with copper foil that contacted the film edges. An infrared thermal imager (UTi160G, Youlide Technology (China) Co., Ltd., Shenzhen, China) was used to measure the temperature of the film. Figure 1 shows a schematic of the process for preparing TCF of AgNWs (Figure 1a) and AgNWs film heater (Figure 1b).

Differential scanning calorimetry (DSC) and thermogravimetric (TG) analysis were conducted via simultaneous differential thermal analysis (STA449F5, NETZSCH-Gertebau GmbH, Selb, Germany). The microstructures of AgNWs films were observed using scanning electron microscope (SEM, Zeiss sigma 500, Carl Zeiss, Jena, Germany), atomic force microscopy (Dimension Edge, Bruker, Billerica, MA, USA), and an optical microscope (Nikon LV100, Nikon Co., Ltd., Tokyo, Japan) with a digital camera. The sheet resistances of films were characterized using a four-point-probe system (ST2253, Suzhou Jingge Electronic Co., Ltd., Suzhou, China) and the optical transmittances were collected by a thin film transmittance meter (GZ502A, Shanghai Guangzhao Photoelectric Technology Co., Ltd., Shanghai, China). The optical transmittance and sheet resistance of each sample were each measured at twenty different sites and calculated from the average value of those measurements. The transmission and diffuse reflectance were measured with PET film as the reference. The surface morphology was analyzed via atomic force microscopy (Dimension Edge, Bruker, Billerica, MA, USA) and six different areas of the surface of sample were selected to obtain root mean square roughness (RMS) value and calculated as average value.

## 3. Results and Discussion

### 3.1. Characterization of Silver Nanowires

To further characterize the AgNWs, we observed the AgNW films via SEM (Figure 2a) and tested their thermal properties (Figure 2b). The inset image in Figure 2a shows a magnified view of the surface. Silver nanowires with diameters of 20–30 nm and lengths of 25–30 μm and nanoparticles can be observed in Figure 2a. As shown in Figure 2b, the DSC curve has three distinct endothermic peaks and one exothermic peak. In combination with the TG analysis curve, the rate of weight loss of the sample in the temperature of the first endothermic peak was as high as 91.50%, which was caused by the volatilization of the isopropanol in the silver nanowire solution. The temperature of the second endothermic peak was about 189.52 °C and the rate of weight loss of the sample was about 92.10%, which was caused by the surfactant decomposing and melting of the silver nanoparticles. Meanwhile, the third endothermic peak was at a temperature of about 332 °C and was caused by the melting of the silver nanostructures. The melting point of the AgNWs was about 197.35 °C, which is far below the melting point of bulk silver (961.78 °C).

### 3.2. Characterization of the Silver Nanowires Films

Figure 3 shows SEM images of AgNWs films on glass slide substrates treated at 25, 100, 150, 170, 200, and 250 °C for 20 min. AgNWs were coated to form random networks by overlapping one another on the substrate surface (Figure 3, 25 °C). After heat-treating the sample at 100 °C for 20 min, the morphology of the AgNWs did not show any obvious change. When the sample was heated at 150 °C, it was found that the size of nanoparticles attached to the AgNWs became slightly larger. Increasing the heat treatment temperature further to 170 °C caused the size of the nanoparticles to increase obviously larger and several notches formed in the AgNWs. Increasing the heat treatment temperature to 200 °C caused the AgNWs to sinter, breaking them into discontinuous segments; this indicates that the temperature of the heat treatment was above the melting point of AgNWs, which accelerated the diffusion of silver atoms at the nanowire surface. We then further increased the temperature of the heat treatment to 250 °C, which caused the AgNWs to fuse into large droplets. Previous reports have also pointed out that high temperature heat treatment induced defects in AgNWs caused by vaporizations, resulting in the emergence of droplets from the AgNWs [31,32,33]. Here, the temperature at which the AgNWs fused into droplets was far lower than the temperatures reported in the literature [34,35]. The same phenomena were also confirmed by optical microscopy (see the Appendix A). The sintering behaviors of nanomaterial are known to be similar to those of the bulk material, including Ostwald ripening, migration, and diffusion of atoms, which can then coalesce elsewhere [35,36,37,38,39,40]. The driving force for surface diffusion is the effective curvature of the free surface of the contact, and the larger is the curvature, the larger is the surface diffusion [40]. The driving force for surface diffusion changes with interface type following: nanoparticle–nanoparticle > nanoparticle–nanowire > nanowire–nanowire [36]. The driving force for neck growth between nanoparticles and interfaces with nanowire is higher than that between nanowire–nanowire interfaces [36]. This is also because atoms are tightly bonded along the nanowire, while the same does not occur at the nanoparticle level as they are “truncated” wires. We observed that the contact interfaces between the nanoparticles and the nanowires melted readily.

The surface topography of the AgNWs films treated at 25, 100, 150, 170, 200, and 250 °C for 20 min were characterized using AFM operated in tapping mode, and the results are shown in Figure 4 (for three-dimensional (3D) images, see Appendix A). The measured root mean square (RMS) roughness values for the corresponding samples were 11.2, 13.4, 14.7, 17.5, 17.9, and 19.1 nm, respectively; these values are all smaller than the diameter of the AgNW. However, it is clear that the RMS film roughness increased along with the temperature of heat treatment. As shown in Figure 3 and Figure 4, possible reasons for this are that the heat-treated nanoparticles and nanowires increased in size or that they formed discontinuous segments or droplets that protruded from the surface. 

### 3.3. Optoelectronic Properties of Silver Nanowires Films

To understand the effects of temperature on the optoelectronic properties of the AgNWs films, we measured the transmittance values (Figure 5a) and sheet resistances (Figure 5b) of films treated at different temperatures for 20 min. Curve a–f in Figure 5a represent the results for films treated at 25, 100, 150, 160, 170, and 180 °C, respectively. The inset in Figure 5a shows the relationship between the transmittance of the film at 550 nm and the heat treatment temperature. The transmittance and the sheet resistance of the films heated at 200 °C are not shown in Figure 5 because the sheet resistance of the film could not be determined, indicating that the film was not conductive. In Figure 5, it can be seen that the transmittance of the films gradually decreased as the temperature of the heat treatment increased. After heating at 25, 100, 150, 160, 170, and 180 °C, the transmittance values at 500 nm were 93.9%, 93.1%, 91.8%, 91.3%, 90.7%, and 87.0%, respectively, and the sheet resistances of the corresponding films were 17.6, 16.5, 14.3, 19.7, 30.1 and 108.1 Ω∙sq^−1^, respectively. Based on the experimental results presented in Figure 2, Figure 3 and Figure 4, we can conclude that the AgNWs heated at 25–150 °C remained intact, and that those AgNW films had high conductivities (17.6–14.3 Ω∙sq^−1^), excellent transmittance values (93.9–91.8%) and low surface roughness values (11.2–14.7 nm); the films therefore have properties that are well-suited for applications in transparent heaters, touch-screen panels, and displays.

### 3.4. Electrothermal Performances of the Silver Nanowires Films

To demonstrate the applicability of the flexible transparent conductive AgNWs film in the field of TFH, we fabricated AgNWs films with sheet resistances of 10, 35, and 130 Ω∙sq^−1^ on PET substrates. The electrothermal performances of the AgNWs film heater were studied by applying direct current to the AgNWs films in a laboratory environment, as schematically illustrated in Figure 1b. Figure 6 shows a plot of temperature versus time for the AgNWs conductive film with a sheet resistance of 10 Ω∙sq^−1^ under the operation for input voltage from 2 to 10 V (Figure 6a) and for different sheet resistances with an input voltage of 10 V (Figure 6b). As shown in Figure 6a, when the input voltages were 5, 7, and 10 V, the electrical powers were 2.5, 4.9, and 10 W, respectively, and the film’s temperatures reached 57, 70, and 110 °C, respectively, confirming that the devices were able to operate with low input voltages. The experimental results indicate that the efficient transduction of electrical energy into Joule heating was caused by the good conductivity of the AgNWs film. Meanwhile, it is worth pointing out that, when the input voltage was in the range of 2–10 V, the film took less than 40 s to reach its steady-state temperature, demonstrating fast response of the AgNWs film heater. The results therefore demonstrate that the film is very suitable for applications in the field of the fast temperature switching with low input voltages. As shown in Figure 6b, when the input voltage was fixed at 10 V, the maximum steady-state temperature achieved increased when the sheet resistance of AgNW film was decreased. When the sheet resistance of the AgNWs film was 10, 35, or 130 Ω∙sq^−1^, the maximum steady-state temperature was 56, 67, or 110 °C, respectively; those values indicate that a sheet resistance of the film in the range of 10–35 Ω∙sq^−1^ is ideal for heating application and that the film exhibited good electrothermal conversion properties. 

In addition, it should pointed out that we measured the transmittance and sheet resistance of corresponding samples (Figure 6a) after being treated at 3, 5, 7, and 10 V. There were no significant changes in the sheet resistance and the transmittance. This result might be an indication that the temperature of the sample was not too high.

Figure 7 shows infrared images of AgNWs film heaters with 10 Ω∙sq^−1^ after being operated at different input voltages for 2 min (Figure 7a) and AgNW films with different sheet resistances under the operation at input voltage of 10 V (Figure 7b). The infrared image in Figure 7a shows low contrast owing to the low temperature generated by the low input voltage (2 V). However, as the input voltage was increased, the temperature of the AgNW film increased, and the infrared images clearly displayed a uniform heat distribution across the film. Conversely, non-uniform AgNWs networks often cause “hot spots”, as can be seen in Figure 7b (130 Ω∙sq^−1^); these hot spots were mainly caused by aggregates of the AgNWs.

To demonstrate the applicability of the AgNW film and the large-area scalability of our process, we fabricated AgNWs film with size of 360 × 270 mm^2^ on a PET substrate. The film was prepared using 3 mL of a AgNWs solution with a concentration of 2.0 mol∙L^−1^ and the Meyer rod was moved at a speed of 0.03 mm/s. We measured the film’s sheet resistance, transmittance, haze, and RMS roughness value. Figure 8 shows a photograph and a 3D AFM image of the sample. We placed a device operating with a blue light emitting diode (LED) on the AgNWs film to measure its transmittance and constructed an AgNWs film heater, as shown Figure 1b. The sheet resistance, transmittance, and haze were found to be 38.6 Ω∙sq^−1^, 92.3%, and 1.16%, respectively, indicating that the sample has excellent optoelectronic properties when applied to PET substrate. The RMS roughness was 16.9, which demonstrates the good surface topography of the sample.

Figure 9 shows the photographs of the LED light on the AgNW film (Figure 9a) and infrared images of the AgNWs film heater (Figure 9b) as well as the relative change in sheet resistance of the film (R and R_0_ represent the sheet resistance of films before and after bending test, respectively) versus the number of conducted bending cycles (Figure 9c). Figure 9 clearly shows that the LED’s light was transmitted through the sample and that the bent conductive film still worked normally. Furthermore, the infrared images of the sample show a uniform heat distribution across the whole surface of the film during both outward and inward bending, indicating that the AgNWs films have a good mechanical flexibility. Figure 9c shows that the relative change in sheet resistance of the film over 300 bending cycles of outward and inward bending was less than 1.5, indicating that the mechanical stability of the AgNW film is insufficient owing to weak adhesion between the AgNWs and the substrate. However, the film still exhibited a good conductivity and heating performance, as shown in Figure 9a,b.

## 4. Conclusions

We fabricated transparent films composed of AgNWs with diameters of 20–30 nm and lengths of 25–30 μm on PET substrates and glass slides via the Meyer rod method. We systematically investigated the AgNW films’ optoelectronic and electrothermal properties. Our experimental results demonstrate that the morphology of the AgNWs showed no significant change when heat-treated at 25–150 °C, and that the AgNW film had a high conductivity (17.6–14.3 Ω∙sq^−1^), excellent transmittance (93.9–91.8%), and low surface roughness (11.2–14.7 nm). We then increased the heat treatment temperature from 170 to 250 °C, which caused the AgNWs to gradually sinter, thereby fusing them together (initially into discontinuous segments and finally into large droplets), which resulted in the conductivity of the film decreasing, until it became non-conductive. We fabricated a transparent AgNW film heater that displayed effective and rapid heating at low input voltages owing to the good conductivity of the AgNW film. With an input voltage in the range of 2–10 V, the film took less than 40 s to reach a steady-state temperature, demonstrating the fast response of the AgNW film heater. When the sheet resistance of the AgNWs film was 10, 35, and 130 Ω∙sq^−1^, the maximum steady-state temperature was 56, 67, and 110 °C, respectively, which indicates good electrothermal conversion behavior of the film for a sheet resistance in the range of 10–35 Ω∙sq^−1^. The conductivity of the film decreased for an increasing number of bending cycles; however, the film still exhibited a good conductivity and heating performances after repeated bending.

## Figures and Tables

**Figure 1 nanomaterials-09-00904-f001:**
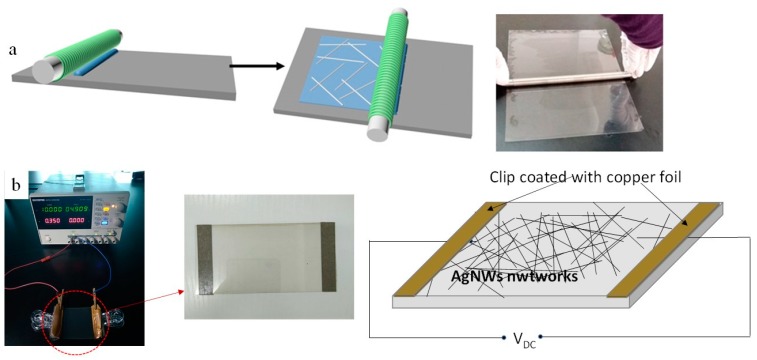
Schematic of the process for preparing: the transparent AgNW film; (**a**) and the AgNWs film heater (**b**).

**Figure 2 nanomaterials-09-00904-f002:**
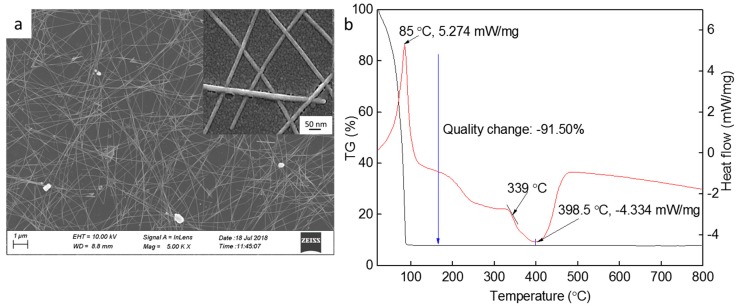
SEM image (**a**); and DSC and TG curves (**b**) of the AgNWs. The inset is the local magnification.

**Figure 3 nanomaterials-09-00904-f003:**
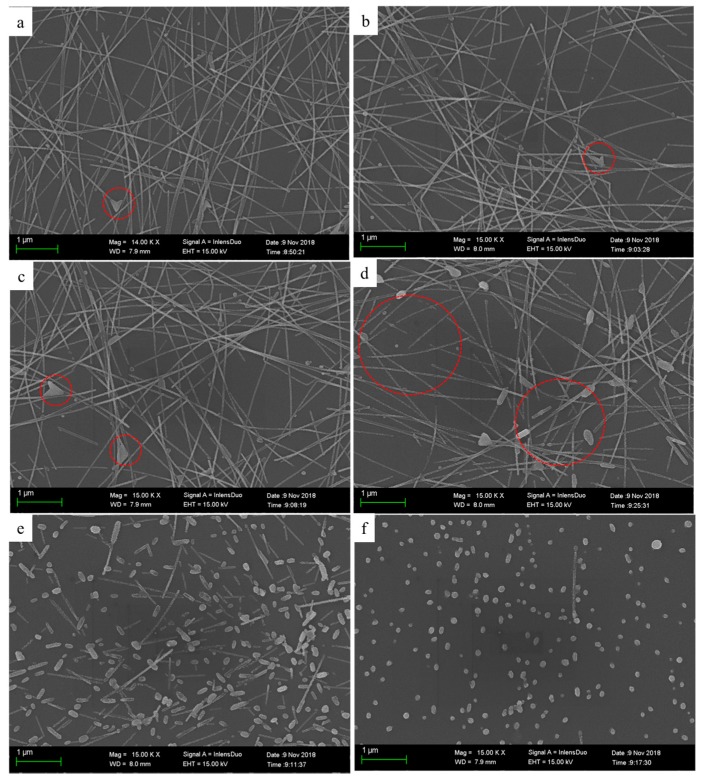
SEM images of AgNWs films treated at different temperatures, (**a**) 25, (**b**) 100, (**c**) 150, (**d**) 170, (**e**) 200, and (**f**) 250 °C for 20 min.

**Figure 4 nanomaterials-09-00904-f004:**
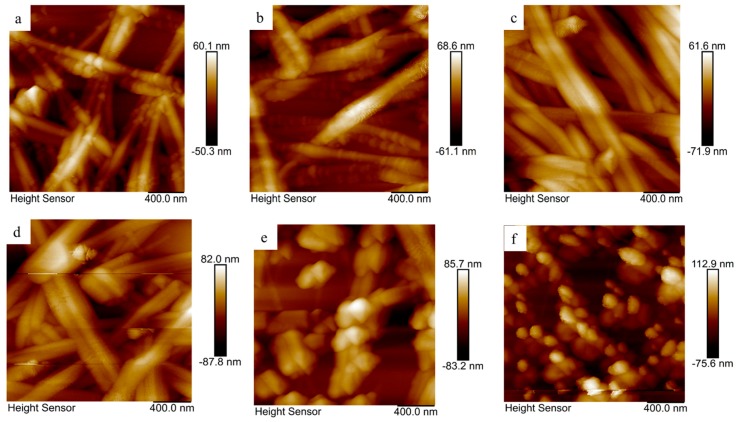
Surface topography of the AgNWs films treated at different temperatures, (**a**) 25, (**b**) 100, (**c**) 150, (**d**) 170, (**e**) 200, and (**f**) 250 °C for 20 min.

**Figure 5 nanomaterials-09-00904-f005:**
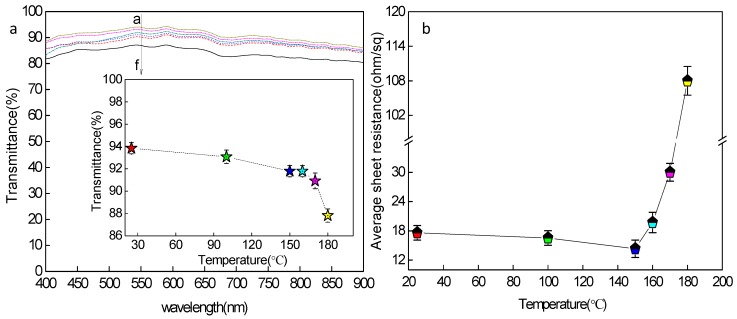
Transmittance (**a**); and sheet resistances (**b**) of films treated at different temperatures for 20 min. Curves a–f in (**a**) represent samples heated at 25, 100, 150, 160, 170, and 180 °C, respectively. The inset in (**a**) shows the relationship between film transmittance at 550 nm and the heat treatment temperature.

**Figure 6 nanomaterials-09-00904-f006:**
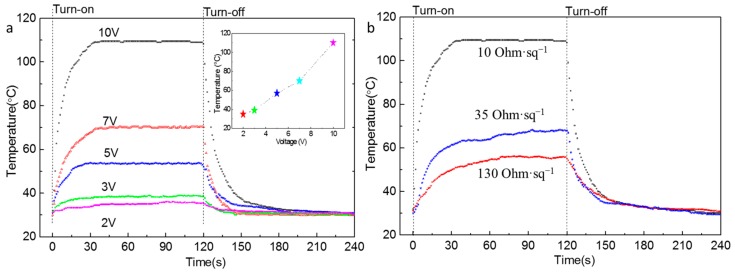
Temperature versus time for: the TFH with the sheet resistance of 10 Ω∙sq^−1^ under operation with different input voltages (**a**); and with different sheet resistances under operation with an input voltage of 10 V (**b**). The inset in (**a**) shows the relationship between the input voltage and the maximum steady-state temperature.

**Figure 7 nanomaterials-09-00904-f007:**
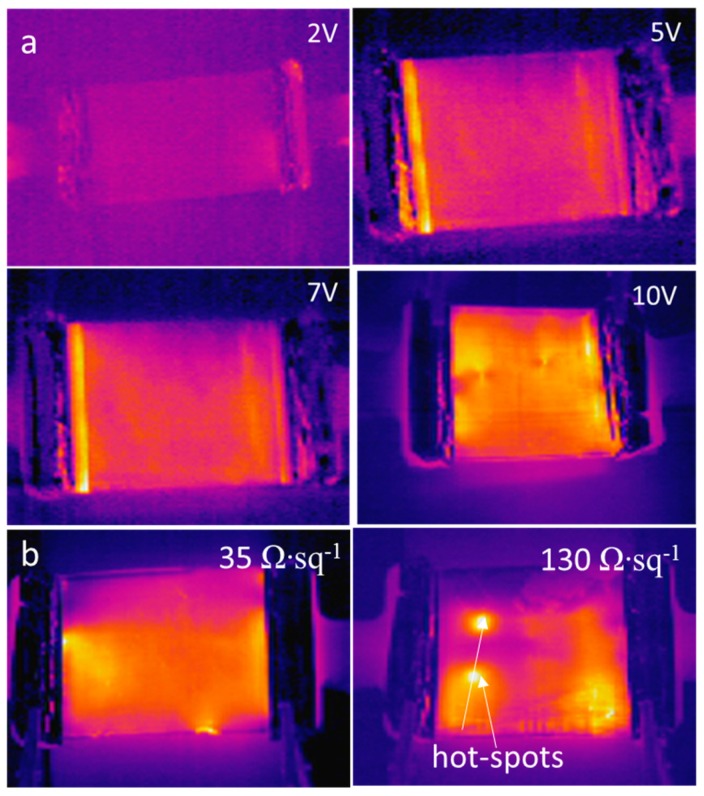
Infrared images of: AgNW film heaters with 10 Ω∙sq^−1^ after operation at different input voltages for 2 min (**a**); and AgNW films with different sheet resistances under operation at input voltage of 10 V (**b**).

**Figure 8 nanomaterials-09-00904-f008:**
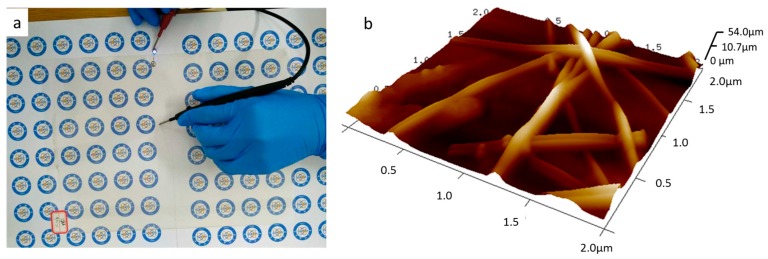
Photograph (**a**); and 3D AFM image (**b**) of the sample.

**Figure 9 nanomaterials-09-00904-f009:**
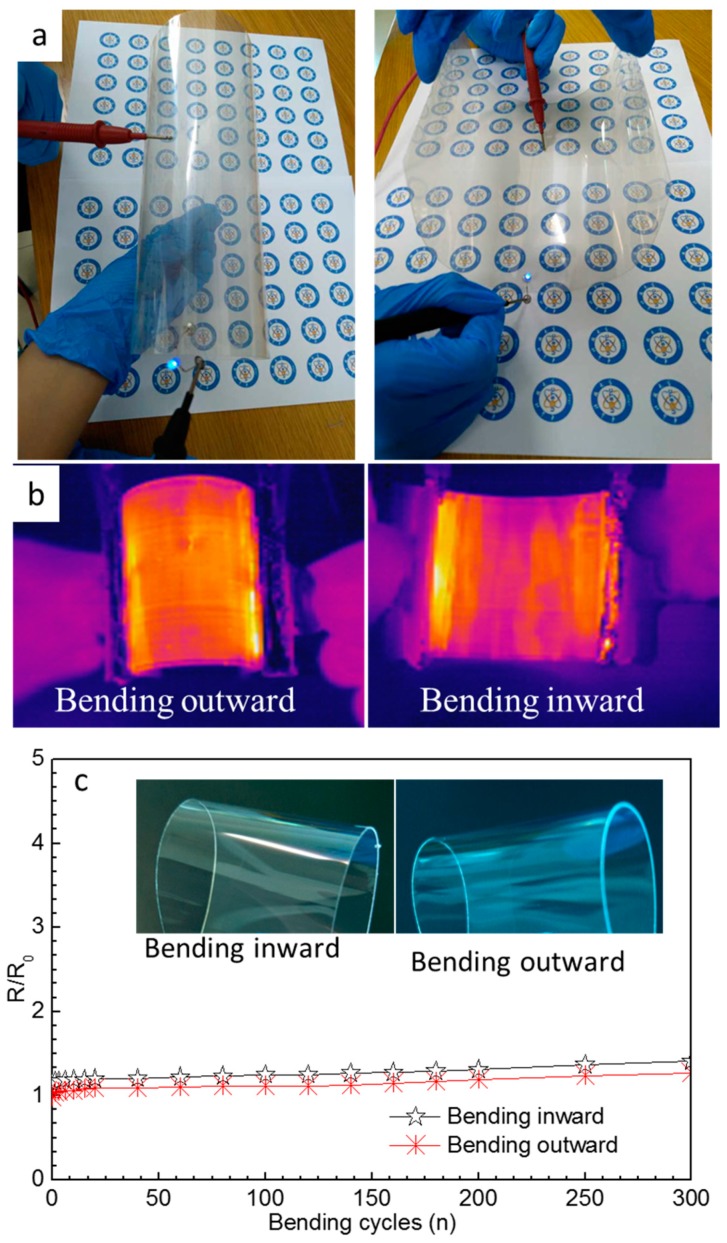
Photographs of an LED placed under the AgNW film during inward and outward bending (**a**); infrared images of the AgNW film heater bent inward and outward (**b**); and the relative change in the sheet resistance of the film versus the number of bending cycles (**c**).

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
