# Peer review of "Optoelectronic and Electrothermal Properties of Transparent Conductive Silver Nanowires Films"

_nanomaterials, 2019, doi:10.3390/nano9060904_

Reviewer 1 Report

Overall this is an excellent paper and research. I have several comments that will help to increase the already high quality of the paper that I recommend the Authors to follow, here listed.

Page 1, line 38

Also make reference to Gallium Zinc Oxide (GZO) as conductive transparent oxide [Plasma Process. Polym. 2015, 12, 725–733]

Line 42 not ITO, better TFH

line 44 not "there", better "those"

Page 2 line 51

add a comment that this is related to the various models for electrical percolation, as discussed in [Nanoscale Research Letters 2014, 9:527] and references there included, and discussed in [Composites: Part A 61 (2014) 108–114]

Line 68 after being cooled

Line 71 by attaching the two ends

Page 3 line 101 Ag NPs melting: could it be related to NW melting across their tiny diameter?

Line 102 Ag Nstructured melting: could it be related to NW melting along their length?

Line 111 where do the NPs come from? raw product?

Line 114 indicated by

Line 120 One clue might be the effective distance of the NWs from the hot plate, please provide the materials used to host the NWs and the thickness that separates them from the plate

Page 4 Line 123 make reference to a relevant paper discussing these results in Ag NPs [Microelectronic Engineering 97 (2012) 8–15]

Line 126 This is also because atoms are tightly bonded along the NWs, while the same does not occurr at the NP level as they are like "truncated" wires. Please add this comment to the text

Lines from 135 to 137: I would not use the RMS. Why? Because especially with NWs there is not a relevant statistics of points. I ask you to use instead (the Maximum height minus the absolute value of Minimum height) divided by two.

Line 139 remove "t"

Page 5 line 149 "inset"

Line 156 "remains intact" and "shows the high"

Figure 4 (a): show legend, besides colors use either dashed lines or different dots to distinguish the curves

Page 6 Figure 5 (a) inset: provide electrical power [W] instead of voltage

Author Response

We appreciate a lot to the reviewer attention and for the reviewer’s evaluation and comments made to the manuscript (nanomaterials-508611). We have revised the manuscript according to your kind advices and reviewer’s detailed suggestion (See revised manuscript with tracked changes). We sincerely hope this manuscript will be finally acceptable to be published on Nanomaterials. Thank you very much for all your help.

Reviewer1:
Overall this is an excellent paper and research. I have several comments that will help to increase the already high quality of the paper that I recommend the Authors to follow, here listed.

Response: Thanks for the reviewer’s kind evaluation and insightful comments. We revised the manuscript according to your kind advice and detailed suggestions. We are trying to work hard and share our work with researchers in the same field.

Action: We revised the manuscript according to reviewer’ kind advice and detailed suggestions.

1)        Page 1, line 38. Also make reference to Gallium Zinc Oxide (GZO) as conductive transparent oxide [Abduev A.; Akmedov A.; Asvarov A.; Chiolerio A. A Revised Growth Model for Transparent Conducting Ga Doped ZnO Films: Improving Crystallinity by Means of Buffer Layers. Plasma Process. Polym. 2015, 12, 725-733]

Response: Thanks for the reviewer’s comments. Ga-doped ZnO thin films are recently raising both scientific and industrial interests due to the lack of natural resources. Indium is a crucial raw material, due to ITO modern touchscreen market. Abduev A. et al revised growth model for transparent conducting Ga doped ZnO films in [Plasma Process. Polym. 2015, 12, 725-733]. This is a very valuable reference, and we are so sorry for ignoring it. We added a reference (Plasma Process. Polym. 2015, 12, 725-733) as Ref. 10 in the revised manuscript.

Action: We added a reference (Plasma Process. Polym. 2015, 12, 725-733) as Ref. 6 in the revised manuscript.

In the revised manuscript:

10.  Abduev A.; Akmedov A.; Asvarov A.; Chiolerio A. A Revised Growth Model for Transparent Conducting Ga Doped ZnO Films: Improving Crystallinity by Means of Buffer Layers. Plasma Process. Polym. 2015, 12, 725-733

2)      Line 42 not ITO, better TFH, line 44 not "there", better "those",

Response: Thanks for the reviewer’s comments. We made mistake for them and just corrected them in the revised manuscript.

Action: We corrected them. Please check them in the revised manuscript by highlighting them in red.

3)        Page 2 line 51, add a comment that this is related to the various models for electrical percolation, as discussed in [Novara C.; Petracca F.; VirgaA.; Rivolo P.; Ferrero S.; Chiolerio A.; Geobaldo F.; Porro S.; Fabrizio Giorgis F. Nanoscale Research Letters, 2014, 9, 527 (1-7)] and references there included, and discussed in [Castellino M.;ChiolerioA.;ShahzadM.I.;Jagdale P.V.;Tagliaferro A. Electrical conductivity phenomena in an epoxy resin-carbon-based materials composite. Composites Part A: Applied Science and Manufacturing.2014, 61,108-114]

Response: Thanks for the reviewer’s comments. We added a comment in the revised manuscript, please check it in line 51 (highlighting in red)

Action: We added a comment in the revised manuscript, please check it in line 51 (by highlighting them in yellow.)

In the revised manuscript:

“……. which is related to the various models for electrical percolation [31, 32].”

4)      Line68 after being cooled; Line 71 by attaching the two ends

Reonse: Thanks for the reviewer’s comments. We made mistakes and corrected them in the revised manuscript. 
Action:  We corrected them in the revised manuscript, please check them.

5)        Page 3 line 101 Ag NPs melting: could it be related to NW melting across their tiny diameter?  Line 102 Ag NW structured melting: could it be related to NW melting along their length?

 Response: Thanks for the reviewer’s insightful comments. The sintering behaviors of nanomaterial include Ostwald ripening, migration and diffusion and merging of atoms, and fusion The driving force for surface diffusion is the effective curvature of the free surface of the contact, and the larger the curvature, the larger the surface diffusion. The driving force for surface diffusion changes with interface type as nanoparticle-nanoparticle > nanoparticles−nanowire > nanowire-nanowire. The driving force of neck growth between the nanoparticles and nanowires interface is higher than that of between the nanowires and nanowires interface. Thus we observed that the contact interfaces between nanoparticles and nanowires are easy to be melted. Maybe Ag NPs melting is related to NW melting across their tiny diameter and Ag NW structured melting is related to NW melting along their length.

Action: Maybe Ag NPs melting is related to NW melting across their tiny diameter and Ag NW structured melting is related to NW melting along their length.

 6)      Line 111 where do the NPs come from? raw product?

Response: Thanks for the reviewer’s comments. Silver nanoparticles come from raw product, it is hard to remove from the silver solution.

Action: Silver nanoparticles come from raw product, it is hard to remove from the silver solution.

 7)      Line 114 indicated by

Response: Thanks for the reviewer’s comments. We corrected them in the revised manuscript.

Action: We corrected them in the revised manuscript. Please check it in the revised manuscript (by highlighting them in yellow.)

8)        Line 120 One clue might be the effective distance of the NWs from the hot plate, please provide the materials used to host the NWs and the thickness that separates them from the plate.

Response: Thanks for the reviewer’s comments. Silver nanowires were deposited on glass slide by Mayer rod, and sample was heat-treated directly on hot plate. Thickness of glass slide is about 1.1±0.04 mm. We did not use anything to separate silver nanowires film from the hot plate.

Action: Silver nanowires were deposited on glass slide by Mayer rod, and sample was heat-treated directly on hot plate. Thickness of glass slide is about 1.1±0.04 mm.

9)        Page 4 Line 123 make reference to a relevant paper discussing these results in Ag NPs [Microelectronic Engineering 97 (2012) 8–15]

Response: Thanks for the reviewer’s comments. We added paper as reference 36.
Action: We added paper as reference 36. Please check it in the revised manuscript.

In the revised manuscript:

36.  Chiolerio A.; Cotto M.; Pandolfi P.;Martino P.; Camarchia, V.; Pirola, M.; Ghione, G. Ag nanoparticle-based inkjet printed planar transmission lines for RF and microwave applications: Considerations on ink composition, nanoparticle size distribution and sintering time. Microelectronic Engineering. 2012 , 97, 8-15

10)    Line 126   This is also because atoms are tightly bonded along the NWs, while the same does not occurr at the NP level as they are like "truncated" wires. Please add this comment to the text.

Response: Thanks for the reviewer’s insightful comments. We added this comment to the text.

Action: We added this comment to the text. Please check it in the revised manuscript.

11)    Lines from 135 to 137: I would not use the RMS. Why? Because especially with NWs there is not a relevant statistics of points. I ask you to use instead (the Maximum height minus the absolute value of Minimum height) divided by two.

Response: Thank you very much for your kind suggestion. Surface roughness is one of the important factors affecting the physical and chemical properties of surfaces. As a representation parameter of surface roughness, RMS cannot accurately depict the surface characteristics of complex structures. We characterized the surface topography of the AgNWs films using AFM (Fig.3 and Fig.S2). Seen from Fig.S2(3D images),it is clearly observe the height of film. For RMS value in the paper, we choose six different areas of the surface of sample to obtain root mean square roughness (RMS) value and calculated as average value. Thank you for your advice to use instead (the maximum height minus the absolute value of Minimum height) divided by two. But I wonder what this value means (the Maximum height minus the absolute value of Minimum height) divided by two. Would you mind giving me some more advice about it? Thanks.

Action: We added some information to explain how to get RMS. Please check them in the revised manuscript. (by highlighting in red)
In the original manuscript:

The surface morphology was analyzed via atomic force microscopy (Dimension Edge, Bruker, America) and six different areas of the surface of sample were selected to obtain root mean square roughness (RMS) value and calculated as average value.

12)  Line 139 remove "t"; Page 5 line 149 "inset"; Line 156 "remains intact" and "shows the high"

Response: Thanks for the reviewer’s comments. We corrected them in the revised manuscript. 
Action:  We corrected them in the revised manuscript. Please check them (by highlighting in red)
13)  Figure 4 (a): show legend, besides colors use either dashed lines or different dots to distinguish the curves

 Response: Thanks for the reviewer’s comments. According to the reviewer’s suggestion, we updated the Fig.4 (a), please check it in the revised manuscript.

Action: We updated the Fig.4 (a), please check it in the revised manuscript.

 In the original manuscript:

 Fig. 4 (a) Transmittance and sheet resistances (b) of films treated at different temperatures for 20 min. Curve a to f are 25 °C, 100 °C,150 °C,160 °C,170 °C, and 180 °C, respectively. The inset in Fig.4a is relationship between the transmittance of film at 550 nm and heat treatment temperature

In the revised manuscript:

Fig. 5 Transmittance (a) and sheet resistances (b) of films treated at different temperatures for 20 min. Curve a to f in panel (a) represent samples heated at 25, 100, 150, 160, 170 °C, and 180 °C, respectively. The inset in Fig.5a shows the relationship between film transmittance at 550 nm and the heat treatment temperature.

 14)  Page 6 Figure 5 (a) inset: provide electrical power [W] instead of voltage

Response: Thanks for the reviewer’s comments. According to the reviewer’s suggestion, we updated the Fig.5 (a), please check it in the revised manuscript.

Action: According to the reviewer’s suggestion, we updated the Fig.4 (a), please check it in the revised manuscript.

In the original manuscript:

Fig.5 Temperature versus time for the TFH with the sheet resistance of 10 Ω∙sq-1 under the operation of different input voltages (a) and with different sheet resistances under the operation of the input voltage of 10 V (b).The inserted in Fig.5a is relationship of the maximum temperature of steady- state with input voltages.

In the revised manuscript:

Fig.6 Temperature versus time for the TFH with the sheet resistance of 10 Ω∙sq-1 under the operation with different input voltages (a) and with different sheet resistances under the operation with an input voltage of 10 V (b).The inset in panel (a)shows the relationship between the input voltage and the maximum steady-state temperature.

Reviewer 2 Report

The paper describes the effect of temperature annealing on the performance of silver nanowire based transparent conductive films. The authors show that up to 150 deg C conditions yield the best optical transparency coupled with sheet resistance. 

In my opinion the paper needs some major revisions on many aspects before it can be published. 

First, I recommend extensive proofreading by a native English speaker as the paper is hard to read as it stands.

Second, the authors have TCH as target application for their material, and claim the novelty lies in the systematic investigation of the effects of temperature on the optical and electrical properties of the films. In my opinion the paper should also investigate the effect of temperature on the TCH performance. Also, temperature effects have been investigated in the past, see for example Kim, ACS Nano 7, 1081 (2013) and Lee, Nano Lett. 8, 689 (2008), which should be cited

One comment I have on the TG analysis is that I would recommend measuring dry samples rather than solutions to avoid observing a 90% mass loss due to the solvent, and to be able to appreciate the rest of the TG curve

As a minor fix, there are two Fig. 1 in the paper.

Author Response

We appreciate a lot to the reviewer attention and for the reviewer’s evaluation and comments made to the manuscript (nanomaterials-508611). We have revised the manuscript according to your kind advices and reviewer’s detailed suggestion (See revised manuscript with tracked changes). We sincerely hope this manuscript will be finally acceptable to be published on Nanomaterials. Thank you very much for all your help.

1)        Reviewer2:

The paper describes the effect of temperature annealing on the performance of silver nanowire based transparent conductive films. The authors show that up to 150 deg C conditions yield the best optical transparency coupled with sheet resistance. In my opinion the paper needs some major revisions on many aspects before it can be published.

Response: Thanks for the reviewer’s attention and evaluation and insightful comments. We revised the manuscript according to your kind advice and trying to work hard and share our work with researchers in the same field.

Action: We revised the manuscript according to your kind advice. Please check them in the revised manuscript.

 1)        First, I recommend extensive proofreading by a native English speaker as the paper is hard to read as it stands.

Response: Thanks for the reviewer’s comments. We corrected some mistakes and added some information in the revised manuscript. In addition, we got help from company who is professional English editing service. Please check them in the revised manuscript.

Action: We corrected some mistakes and added some information in the revised manuscript. In addition, we got help from company who is professional English editing service. Please check them in the revised manuscript.

2)        Second, the authors have TCH as target application for their material, and claim the novelty lies in the systematic investigation of the effects of temperature on the optical and electrical properties of the films. In my opinion the paper should also investigate the effect of temperature on the TCH performance. Also, temperature effects have been investigated in the past, see for example Kim, ACS Nano 7, 1081 (2013) and Lee, Nano Lett. 8, 689 (2008), which should be cited.

Response: Thanks for the reviewer’s comments. TCH usually needs to be connected to the circuit, and the temperature of film increases at a certain input voltage, so we can obtain the relationship of the temperature with the TCH performance, just as shown in Fig.5. In fact, when TCF is used as electrode, the effect of temperature on the film performance, just as shown in Fig.4. It is difficult to test the transmittance and sheet resistance of samples when the circuit is switched on.  Kim et al (Kim A.; Won Y.; Woo K.; Kim C-H.; Moon J. Highly Transparent Low Resistance ZnO/Ag Nanowire/ZnO Composite Electrode for Thin Film Solar Cells. ACS Nano. 2013, 2: 1081-1091) reported the variations of the resistance of the AgNWs single film and the ZnO/Ag/ZnO composite film on glass substrates as a function of the annealing temperature (Fig.5), same as Fig.4 in our paper.  Lee et al (Lee J-Y.; Connor S. T.; Yi Cui Y.; Peumans P. Solution-Processed Metal Nanowire Mesh Transparent Electrodes. Nano Lett. 2008, 8, 689-692) reported sheet resistance vs annealing time for an Ag nanowires mesh annealed at 200 ºC (Fig.3).

   In our work, we measured the transmittance and sheet resistance of corresponding samples (Fig.5 in the original manuscript) after treated at input voltages. We did not observe difference in both of the sheet resistance and the transmittance of samples after treated different input voltages. The average sheet resistance is about 10 Ω∙sq-1, and transmittance as following. It is maybe that the temperature of sample was not too high. Because there is no significant change in the sheet resistance and transmittance of sample, so we did not show it in the paper.

Action: We added some sentences in the revised manuscript, please check them.

In the revised manuscript:

In addition, it should point out that we measured the transmittance and sheet resistance of corresponding samples (Fig.5a) after treated at 3V, 5 V, 7 V, and 10 V. There are no significant change in the sheet resistance and the transmittance. The result might be that the temperature of sample was not too high.

3)        One comment I have on the TG analysis is that I would recommend measuring dry samples rather than solutions to avoid observing a 90% mass loss due to the solvent, and to be able to appreciate the rest.

Response: Thank you very much for your kind suggestion. It is good idea to measure TG with dry samples. We considered measuring TG with dry sample, but because the nanowires were serious agglomerated after drying,we worried that there would be a big error, so we chose solution to measure TG.

Action: We deeply appreciate the reviewer’s insightful comments and kind advice. We considered measuring TG with dry sample, but there would be some error due to serious agglomerates of dried silver nanowires, so we used solution to analyze on the TG.

 4)      As a minor fix, there are two Fig. 1 in the paper.

 Response: Thank you very much for your kind suggestion. We made mistake and corrected it in the revised manuscript.

 Action: We corrected it in the revised manuscript. Please check them in the revised manuscript by highlighting in red.

Reviewer 3 Report

The authors present interesting results based on the AgNW. I have a few comments:

1)     Th authors mentioned that the transmittance and sheet resistance measurements were performed at twenty different sites and were calculated as average values. In my opinion the results should be presented statistically (the value of T@550nm and Rsq. should be presented with uncertainty).

2)     The authors mentioned that that the heat-treated nanoparticles and nanowires become large or form to the discontinuous segments or droplets, which protruded from the surface. This effect is associated with the temperature of treatment and can be treated as a degradation of a layer. Were these measurements performed for longer annealing times (especially for lower temperatures 25, 200 and 150C)? These measurements seems to be important from the practical point of view – durability of devices.

Author Response

We appreciate a lot to the reviewer attention and for the reviewer’s evaluation and comments made to the manuscript (nanomaterials-508611). We have revised the manuscript according to your kind advices and reviewer’s detailed suggestion (See revised manuscript with tracked changes). We sincerely hope this manuscript will be finally acceptable to be published on Nanomaterials. Thank you very much for all your help.

Reviewer 3:

The authors present interesting results based on the AgNW. I have a few comments:

Response: Thanks for the reviewer’s attention and evaluation and insightful comments. We revised the manuscript according to your kind advice and trying to work hard and share our work with researchers in the same field.

Action: We revised the manuscript according to your kind advice. Please check them in the revised manuscript.

1)        The authors mentioned that the transmittance and sheet resistance measurements were performed at twenty different sites and were calculated as average values. In my opinion the results should be presented statistically (the value of T@550nm and Rsq. should be presented with uncertainty).

Response: Thanks for the reviewer’s attention and evaluation and insightful comments. Yes, for the transmittance and sheet resistance measurements of film, the statistical date are relatively accurate due to the ununiformity of distribution of silver nanowires. A better way is to select several samples, and the transmittance and sheet resistance measurements were performed at twenty different sites on each sample and calculated the transmittance at 550 nm and sheet resistance as average value.However, due to the unrepeatable silver nanowires network structure, it is still statistically reasonable to perform the transmittance and sheet resistance at 20 different points on the surface of a sample. In additon, in order to match the transmission spectrum with the sheet resistance and transmittance at 550 nm, we think it is better to keep data and figure as Figure 5.

Action: We have not modified the data. It is still statistically reasonable to perform the transmittance and sheet resistance at 20 different points on the surface of a sample due to the unrepeatable silver nanowires network structure.   

2) The authors mentioned that that the heat-treated nanoparticles and nanowires become large or form to the discontinuous segments or droplets, which protruded from the surface. This effect is associated with the temperature of treatment and can be treated as a degradation of a layer. Were these measurements performed for longer annealing times (especially for lower temperatures 25, 200 and 150C)? These measurements seems to be important from the practical point of view – durability of devices

Response: Thanks for the reviewer’s comments. All samples were air dried at room temperature for 2 min and then treated on digital hotplate at temperature from 25 °C to 250 °C for 20 min.
Action: All samples were air dried at room temperature for 2 min and then treated on digital hotplate at temperature from 25 °C to 250 °C for 20 min.

Round  2

Reviewer 2 Report

I am satisfied with authors' responses. 

Please only note that in line 102 you are discussing thermal rather than thermodynamic properties.

I thus recommend publication of this paper in the journal Nanomaterials

Author Response

Response to the reviewers’ report Thank you very much for reviewer’s the attention and insightful comments made to the manuscript (nanomaterials-508611). We have revised the manuscript according to your kind advices and reviewer’s detailed suggestion (See revised manuscript with tracked changes). We sincerely hope this manuscript will be finally acceptable to be published on Nanomaterials. Thanks again. 1) Reviewer2: I am satisfied with authors' responses. Please only note that in line 102 you are discussing thermal rather than thermodynamic properties. I thus recommend publication of this paper in the journal Nanomaterials Response: Thanks for the reviewer’s kind comments and suggestion. We corrected the word “ thermodynamic” to “thermal” that in line 102. Action: We corrected the word “thermodynamic” to “thermal” that in line 102.Please check it in the revised manuscript by highlighting in red.

Reviewer 3 Report

I agree with the authors' statement:

"Yes, for the transmittance and sheet resistance measurements of film, the statistical date are relatively accurate due to the ununiformity of distribution of silver nanowires. A better way is to select several samples, and the transmittance and sheet resistance measurements were performed at twenty different sites on each sample and calculated the transmittance at 550 nm and sheet resistance as average value."

However, I am not convinced by the authors' explanation regarding the values of T for 550 nm and sheet resistance. In my opinion uncertainties for the T@550 nm and the sheet resistance should be added.

This information can be helpful for readers - gives information about variations between samples.

Author Response

Thank you very much for your attention and insightful comments made to the manuscript (nanomaterials-508611). We have revised the manuscript according to your kind advices (See revised manuscript with tracked changes). We sincerely hope this manuscript will be finally acceptable to be published on Nanomaterials.

Thank you agian for all your help.

Reviewer 3:

I agree with the authors' statement:

"Yes, for the transmittance and sheet resistance measurements of film, the statistical date are relatively accurate due to the ununiformity of distribution of silver nanowires. A better way is to select several samples, and the transmittance and sheet resistance measurements were performed at twenty different sites on each sample and calculated the transmittance at 550 nm and sheet resistance as average value."

However, I am not convinced by the authors' explanation regarding the values of T for 550 nm and sheet resistance. In my opinion uncertainties for the T@550 nm and the sheet resistance should be added.

This information can be helpful for readers - gives information about variations between samples.

Response: Thanks for the reviewer’s insightful comments. Yes, the information of the uncertainties of the T@550 nm and the sheet resistance is helpful for readers. According to the reviewer’s comment, we updated the Fig.5 and added error bar in Fig.5, please check them in the revised manuscript.

Action: We updated the Fig.5 and added error bar in Fig.5, please check them in the revised manuscript.

Fig. 5 Transmittance (a) and sheet resistances (b) of films treated at different temperatures for 20 min. Curve a to f in panel (a) represent samples heated at 25, 100, 150, 160, 170 °C, and 180 °C, respectively. The inset in Fig.5a shows the relationship between film transmittance at 550 nm and the heat treatment temperature.

Round  3

Reviewer 2 Report

The paper can be published

Reviewer 3 Report

The Authors completed the manuscript.